# AUTOMATIC CURRICULUM FOR UNSUPERVISED REINFORCEMENT LEARNING

## ABSTRACT

Recent unsupervised reinforcement learning (URL) can learn meaningful skills without task rewards by carefully designed training objectives. However, most existing works lack quantitative evaluation metrics for URL but mainly rely on visualizations of trajectories to compare the performance. Moreover, each URL method only focuses on a single training objective, which can hinder further learning progress and the development of new skills. To bridge these gaps, we first propose multiple evaluation metrics for URL that can cover different preferred properties. We show that balancing these metrics leads to what a "good" trajectory visualization embodies. Next, we use these metrics to develop an automatic curriculum that can change the URL objective across different learning stages in order to improve and balance all metrics. Specifically, we apply a non-stationary multi-armed bandit algorithm to select an existing URL objective for each episode according to the metrics evaluated in previous episodes. Extensive experiments in different environments demonstrate the advantages of our method on achieving promising and balanced performance over all URL metrics.

## 1 INTRODUCTION

Reinforcement learning (RL) has recently achieved remarkable success in autonomous control (Kiumarsi et al., 2017) and video games (Mnih et al., 2013). Its mastery of Go (Silver et al., 2016) and large-scale multiplayer video games (Vinyals et al., 2019) has drawn growing attention. However, a primary limitation for the current RL is that it is highly task-specific and easily overfitting to the training task, while it is still challenging to gain fundamental skills generalizable to different tasks. Moreover, due to the sparse rewards in many tasks and poor exploration of the state-action space, RL can be highly inefficient. To overcome these weaknesses, intrinsic motivations (Oudeyer & Kaplan, 2009) have been studied to help pre-train RL agents in earlier stages even without any task assigned. The so called "unsupervised RL (URL)" does not rely on any extrinsic task rewards and its primary goal is to encourage exploration and develop versatile skills that can be adapted to downstream tasks.

Although URL provides additional objectives and rewards to train fundamental and task-agnostic skills, it lacks quantitative evaluation metrics and yet mainly relies on visualizations of trajectories to demonstrate its effectiveness. Although it can be evaluated through downstream tasks by their extrinsic rewards (Laskin et al., 2021b), this requires further training and can be prone to overfitting or bias towards specific tasks. A key challenge in developing evaluation metrics for URL is how to cover different expectations or preferable properties for the agent, which usually cannot be all captured by a single metric. Recently, IBOL (Kim et al., 2021) introduced the concept of disentanglement to evaluate the informativeness and separability of learned skills. However, it does not consider other characteristics such as the coverage over the state space. In addition, how to balance multiple metrics in the evaluation is an open challenge. Therefore, it is critical to develop a set of metrics that can provide a complete and precise evaluation of an URL agent.

In this paper, we take a first step towards quantitative evaluations of URL by proposing a set of evaluation metrics that can cover different preferred capabilities of URL, e.g., on both exploration and skill discovery. In case studies, we show that URL achieving balanced and high scores over all the proposed metrics fulfills our requirements for a promising pre-trained agent. However, excelling on only one metric cannot exclude certain poorly learned URL policies.

In contrast to the ambiguity of evaluation metrics for current URL, the existing intrinsic rewards for URL are quite specific and focused, e.g., the novelty/uncertainty of states (Pathak et al., 2017; Burda et al., 2019; Pathak et al., 2019), the entropy of state distribution (Lee et al., 2019; Mutti et al., 2021; Liu & Abbeel, 2021a), and the mutual information between states and skills (Gregor et al., 2017; Eysenbach et al., 2019b), which are task-free and can provide dense feedback. For most URL methods, each one only focuses on learning with a single intrinsic reward. They mainly differ on implementations, e.g., how to define the novelty, how to estimate the state entropy or mutual information. However, the quality of implementations significantly depends on the modeling of the environment dynamics, which cannot be always accurate everywhere if the exploration is only guided by a single reward. For example, as shown later, agents learning with a single intrinsic reward for exploration could be hindered from further exploration since its novelty approximation is limited to local regions. Moreover, in order to achieve consistent improvement on multiple evaluation metrics and balance their trade-offs, training with a single intrinsic reward is not enough. Hence, it is necessary in URL to take multiple intrinsic rewards into account.

In this paper, we leverage multiple existing intrinsic rewards and aim to automatically choose the most helpful one in each learning stage for optimizing the proposed multiple evaluation metrics. This produces a curriculum of URL whose training objective is adjusted over the course of training to keep improving all evaluation metrics. Since the intrinsic reward is varying concurrently with URL on the fly, we apply a multi-objective multi-armed bandits algorithm to address the exploration-exploitation trade-off, i.e., we intend to select the intrinsic reward (1) that has been rarely selected before (exploration) or (2) that results in the greatest and balanced improvement over all the metrics in history (exploitation). Specifically, we adopt Pareto UCB (Drugan & Nowe, 2013) to optimize the multi-objective defined by the metrics and then extend it to capture the non-stationary dynamics of curriculum learning, i.e., the best intrinsic reward may change across learning stages. This assumption is in line with our observation that a single intrinsic reward cannot keep improving all metrics and URL may stop exploration and ends with sub-optimal skills.

To the best of our knowledge, our work is among a few pioneering studies focusing on developing evaluation metrics for URL. While automatic curriculum learning (ACL) has achieved success in deep RL (Portelas et al., 2020), it has not been studied for URL, though adaptively changing intrinsic motivations is a natural human learning strategy in exploring an unknown world.In experiments, we evaluate our approach on challenging URL environments. Our method consistently achieves better and more balanced results over multiple evaluation metrics than SOTA URL methods. Moreover, we present thorough empirical analyses to demonstrate the advantages brought by the automatic curriculum and the multi-objective for optimizing the curriculum.

## 2 RELATED WORKS

**Unsupervised Reinforcement Learning**. Intrinsic rewards are used for training URL. For exploration, intrinsic motivations can be based on curiosity and surprise of environtal dynamics (Di Domenico & Ryan, 2017), such as Intrinsic Curiosity Module (ICM) (Pathak et al., 2017), Random Network Distillation (RND) (Burda et al., 2019), and Disagreement (Pathak et al., 2019). Another common way to explore is to maximize the state entropy. State Marginal Matching (SMM) (Lee et al., 2019) approximates the state marginal distribution, and matching it to the uniform distribution is equivalent to maximizing the state entropy. Other methods approximate state entropy by particle-based method MEPOL (Mutti et al., 2020), APT (Liu & Abbeel, 2021a), ProtoRL (Yarats et al., 2021), APS (Liu & Abbeel, 2021b). Mutual information-based approaches have been used for self-supervised skill discovery, such as VIC (Gregor et al., 2017), DIAYN (Eysenbach et al., 2019a), VALOR (Achiam et al., 2018). VISR (Hansen et al., 2020) also optimizes the same ojective, but its special approximation brought successor feature (Barreto et al., 2016) into unsupervised skill learning paradigm and enables fast task inference. APS (Liu & Abbeel, 2021b) combines the exploration of APT and successor feature of VISR.

**Automatic Curriculum Learning**. Automatic curriculum learning has been widely studied. It allows models to learn in a specific order for learning harder tasks more efficiently (Graves et al., 2017; Bengio et al., 2009). In RL, a lot of work considers scheduling learning tasks (Florensa et al., 2018; 2017; Fang et al., 2019; Matiisen et al., 2019; Schmidhuber, 2013). In URL, handcrafted curriculum is used by EDL (Campos et al., 2020) and IBOL (Kim et al., 2021). EDL first explores, then assigns

the discovered states to skills, and finally learns to achieve those skills. IBOL also explores and assigns skill after a specific linearzer learning phase. Both of them are not automatic curriculum, and the number of training steps for each training phase needs to be specified before training. In addition, VALOR (Achiam et al., 2018) mentioned curriculum learning, but their curriculum is just gradually increasing the number of skills.

## 3 PRELIMINARIES

**MDP without external rewards** In Markov Decision Process (MDP) $\mathcal{M} = (\mathcal{S}, \mathcal{A}, p)$ *without external rewards*, $\mathcal{S}$ and $\mathcal{A}$ respectively denote the state and action spaces, and $p(s_{t+1}|s_t, a_t)$ is the transition function where $s_t, s_{t+1} \in \mathcal{S}$ and $a_t \in \mathcal{A}$. Given a policy $\pi(a_t|s_t)$, a trajectory $\tau = (s_0, a_0, \ldots, s_T)$ follows the distribution $\tau \sim p(\tau) = p(s_0) \prod_{t=0}^{T-1} \pi(a_t|s_t) p(s_{t+1}|s_t, a_t)$. We formulate the problem of unsupervised skill discovery as learning a skill-conditioned policy $\pi(a_t|s_t, z)$ where $z \in \mathcal{Z}$ represents the latent skill. The latent representations of skills $z$ can be either continuous $z \in \mathbb{R}^d$ or discrete $z \in \{z_1, z_2, ..., z_{N_z}\}$. $H(\cdot)$ and $I(\cdot; \cdot)$ denote entropy and mutual information, respectively.

**Intrinsic Rewards for URL** Intrinsic Curiosity Module (ICM) (Pathak et al., 2017) defines the intrinsic reward as the error between the state prediction of a learned dynamics model and the observation. Not good at handling aleatoric uncertainties in the dynamics: $r(s, a, s') \propto ||g(s'|s, a) - s'||^2$.

Disagreement (Pathak et al., 2019) is similar to ICM but instead trains an ensemble of forward models and defines the intrinsic reward as the variance (or disagreement) among the models. Better at handling aleatoric uncertainties since high stochasticity in the environment will result high prediction error but low variance if it has been thoroughly explored. It is defined as $r(s, a) \propto \mathrm{Var}\{g_i(s_{t+1}|s_t, a_t)\},\ i = 1, ..., N$.

State Marginal Matching (SMM) (Lee et al., 2019) learns a parametric density function $q_\theta$ to approximate state distribution. It tries to match the state distribution to a uniform prior $p^*$ that maximizes $H(S)$: $r(s) = \log p^*(s) - \log q_\theta(s)$.

Active Pre-training (APT) (Liu & Abbeel, 2021a) utilizes a particle-based estimator that uses K nearest-neighbors to estimate state entropy in a non-parametric way. It is defined as $r(s) \propto \sum_{j \in \mathrm{random}} \log \|s_t - s_j\|\quad j = 1, \ldots, K$.

DIYAN (Eysenbach et al., 2019b) optimizes the mutual information between states and skills. It learns a parametric disciminator $q_\theta$, and optimize $-H(Z|S)$ by the intrinsic reward: $r(s, z) \propto \log q_\theta(z|s) + \mathrm{const}$.

## 4 METHODOLOGY

### 4.1 OVERVIEW OF AUTOMATIC CURRICULUM FOR URL

Instead of keep using one intrinsic reward for training an URL agent, we allow the agent choosing one reward among multiple candidates in each learning stage for improving multiple evaluation metrics. This generates an automatic curriculum for URL whose goal is to find a sequence of intrinsic rewards that optimizes multi-objectives each corresponding a metric. The framework of our proposed automatic curriculum method is illustrated in Fig 1. Our curriculum adds an outer loop outside the conventional URL framework (i.e., the interaction between RL agent and environment). The curriculum has a reward selection module that selects an intrinsic reward for each learning stage based on multiple evaluation metrics computed on the replay buffer. By allocating the intrinsic reward that can result in the greatest improvement on multiple metrics in each stage, the curriculum aims to find an optimal sesquence of rewards to keep improving the URL loop and optimize all the metrics.

Given previous work in URL, we still need to address two primary new challenges in building the curriculum: (1) what are the evaluation metrics? and (2) how to select the intrinsic reward for each learning stage? We propose our solutions to these two problems in Section 4.2 and 4.3, respectively. In Section 4.2, we propose multiple metrics to evaluate the capability of URL on exploration and

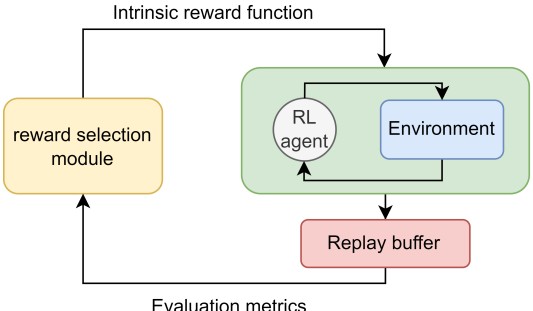

Figure 1: Block diagram of our proposed method

skill learning. These metrics not only include existing ones but also cover other preferred properties. In Section 4.3, we discuss the exploration-exploitation trade-off for award selection in URL and extend an multi-objective multi-armed bandits algorithm to make non-stationary decisions on the reward used for each learning stage's URL.

## 4.2 MULTIPLE EVALUATION METRICS

In the following of this section, Our goal developing general and consistent metrics to evaluate the process of exploration and skill learning. We start to describe our methods with toy examples.

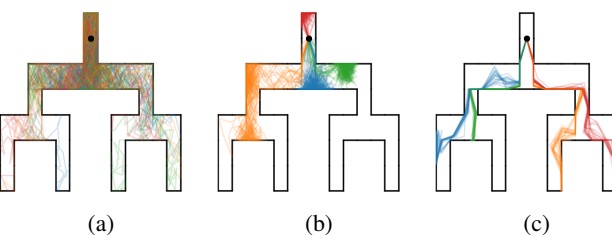

(a)            (b)            (c)

Figure 2: Examples of tree maze trajectories

In Fig. 2, the 3 subfigures show the trajectories of 3 different agents: (a) an unconverged agent, (b) an APT agent and (c) an agent by our proposed method. Each agent has 4 different skills. Each skill has trajectories with its own color. Although the trajectories in Fig. 2a seem to have covered more state space than others, the agent's skill is not well learned. Trajectories of different skills in Fig. 2b are more separable but they not well balanced because the yellow skill covers more states than others. Fig. 2c seems to have explored the state space well and learned balanced skills.

Previous works rely on visualizations of trajectories to compare the performance of exploration and skill learning (Campos et al., 2020) (Kim et al., 2021) (Park et al., 2022) (Eysenbach et al., 2019b). By visualization, we might consider the agent in Fig. 2c to be the best overall, but we do not have a quantitative reason for it. Next, we will define 3 general metrics that quantitatively evaluate exploration and skill learning. And the evaluation results using the proposed metrics for Fig. 2 can be found in Section 5.2.

**State Coverage (SC)** This metric evaluates how much of the state space the agent can cover. Similar to previous methods that approximate the state entropy using particle-based methods, our metric is also based on particle-based entropy. By (Singh et al., 2003) the particle-based entropy estimation should be a sum of the log of the distance between each particle and its $k$-th nearest neighbor, defined as

$$H_{\text{PB}}(S) \propto \sum_{i=1}^{n} \log \|s_i - s_i^{(k)}\|. \tag{1}$$

For robust and stable implementation, we use the modified version from APT (Liu & Abbeel, 2021a), see Appendix A. We find that with large numbers of $n$ and $k$, this estimation can reflect what a good state coverage is in visualization, so we apply it for a large number of recent states in the buffer to evaluate the learning progress of state coverage.

*Remark.* Our purpose is to have an automatic curriculum that trying to optimize evaluation metrics by selecting intrinsic reward. For this state coverage metric, one question could be why not use itself as the intrinsic reward and directly optimize it with RL algorithms as APT did. The main reason is that APT agent tend to expand its trajectory locally instead of reaching to new areas. Combining it with other intrinsic rewards help to escape this local optimum. In fact, the agent in Fig. 2b is an APT agent and the one in Fig. 2c learns by a curriculum. What's more, we find it expensive to estimate with large $n$ and $k$ for every RL update, so the entropy estimation in APT could be less accurate.

**Particle-based Mutual Information (PMI)**    Mutual information between state $S$ and a latent skill $Z$ is an essential objective for URL to learn a skill conditioned agent. Intuitively, Eysenbach et al. (2021) showed that, under some assumption, maximizing the objective $I(S; Z)$ initialize the agent to be optimal for certain downstream tasks. Mutual information between state and skill can be expanded as

$$I(S; Z) = H(S) - H(S|Z). \tag{2}$$

It is one entropy subtracting another, so we propose to implement particle-based entropy to approximate them and obtain the mutual information.

$$I(S; Z) \approx \hat{H}_{\mathrm{PB}}(S) - \hat{H}_{\mathrm{PB}}(S|Z). \tag{3}$$

The approximation bias of particle-based entropy in Eq. 1 depends on $k$ and $n$, so with the large sample numbers of $n$ and $k$ for both $\hat{H}_{\mathrm{PB}}(S)$ and $\hat{H}_{\mathrm{PB}}(S|Z)$, we can get accurate approximation from this substraction.

**Mutual information for individual skill**    When optimizing the mutual information objective, it is possible to learn some skills that are less informative than others. So it is necessary to evaluate the informativeness of an individual skill. There have not been any of this kind of metrics in URL yet. To fill this gap, we define a novel metric as $I(S; \mathbf{1}_z)$, where $\mathbf{1}_z$ is the indicator of $Z = z$ and is a random variable. The estimation of this metric is shown in Appendix B.

We find that $I(S; \mathbf{1}_z)$ measures not only the informativeness of this skill, but also the separability. In the URL skill learning setting, separability between one skill and others means that the states inferred by this skill should share little overlaps with the states inferred by others. Therefore, it means that the states should be certain to be inferred by one single skill, thus meaning low $H(\mathbf{1}_z|S)$.

Because $I(S; \mathbf{1}_z) = H(S) - H(S|\mathbf{1}_z)$, for situations with the same state and skill distributions, the one with larger $I(S; \mathbf{1}_z)$ would be lower in $H(S|\mathbf{1}_z)$. since $H(\mathbf{1}_z|S) = H(S|\mathbf{1}_z) + H(\mathbf{1}_z) - H(S)$, lower $H(S|\mathbf{1}_z)$ means also low in $H(\mathbf{1}_z|S)$ thus better separability.

**Disentanglement**    IBOL (Kim et al., 2021) first introduced URL with the concept of disentanglement, a motivation for this metric from a theoretical perspective is in Appendix. C. This concept includes two aspects: Informativeness and separability. Informativeness here means information shared between a skill and its inferred states. Separability in representation means that there should be no information shared between two latent dimensions, while here it means the trajectories inferred by two different skills should be separated from each other.

They evaluate disentanglement with metrics such as SEPIN@k and WSEPIN (Do & Tran, 2019) to measure disentanglement. SEPIN@k is the top k average of $I(S, Z_i|Z_{\neq i})$ and WSEPIN is the average of $I(S, Z_i|Z_{\neq i})$ weighed on $I(S, Z_i)$.

However, because SEPIN@k and WSEPIN were originally used for representation learning, $Z_i$ was meant to refer to independent representations. For skill-conditioned RL, skill $Z$ is a single random variable, so for URL setting, it is not suitable to directly use $I(S, Z_i|Z_{\neq i})$, which requires $Z_i$ and $Z_{\neq i}$ to be two individual random variables.

With $I(S; \mathbf{1}_z)$ to measure the informativeness and separability of skill $z$, we can define metrics for the disentanglement of learned skills. Again, because SEPIN@k and WSEPIN are defined for

representation learning, where the more informative representations are valued, while the lesser ones are ignored, how SEPIN@k and WSEPIN average over the representations is not suitable for the skills in the URL setting. Instead of ignoring the ones with lower informativeness, all learned skills of an URL agent have an impact on the agent's behavior, so we care about the median and minimum of the informativeness of the skills.

We defined Median SEParability and INformativeness (MSEPIN) as

$$\text{MSEPIN} = \underset{z}{\text{med}}\, I(S; \mathbf{1}_z), \tag{4}$$

where med is the median over skills $z$, and Least SEParability and INformativeness (LSEPIN) as
$z$

$$\text{LSEPIN} = \min_z I(S; \mathbf{1}_z). \tag{5}$$

### 4.3 Automatic Curriculum for URL

Existing URL methods only use a single intrinsic reward to train their policies. Since the accuracy of the reward depends on the agent's modeling of the environment dynamics, whose quality heavily relies on the data collected through the agent's exploration, URL keeping using the same reward may stop exploration earlier with a sub-optimal policy. In this paper, we instead use multiple intrinsic rewards because different rewards can benefit the learning process from different perspectives. This is shown in Section 5.3, where a curriculum combining two intrinsic rewards made them complementary to each other and thus performs better than a single-reward URL. Moreover, it also showed that a random sequence of intrinsic rewards could not be better than a single one, so the order of intrinsic rewards matters in URL. Therefore, our goal is to find a curriculum, i.e., a sequence of intrinsic rewards selected from a candidate set.

The curriculum should also be unsupervised, so no prior knowledge or extrinsic metric is allowed. the mechanism of choosing the next intrinsic reward for the agent should only be based on the historical information collected from environmental interactions. In order to make better choices, it also needs to try different intrinsic rewards and evaluate the improvement they bring to URL. Therefore, an exploration-exploitation trade-off process, e.g., a multi-armed bandit algorithm, is critical to the curriculum development.

Since our goal is an agent excelling on multiple evaluation metrics, the curriculum should take all these multiple objectives into account when selecting a reward for the next stage training. Hence, we formulate it as a Multi-objective multi-armed bandit problem and adopt empirical Pareto UCB (Drugan & Nowe, 2013) algorithm because of its easier implementation and tighter regret bound. In addition, due to the non-stationary dynamics of a curriculum, the best intrinsic reward may change across learning stages and we need a non-stationary extension of Pareto UCB to capture the changes.

We consider automatic curriculum learning based on Empirical Pareto UCB to learn a task selection function $\mathcal{D} : \mathcal{H} \to \mathcal{U}$ where $\mathcal{H}$ can contain any information about previous interactions and $\mathcal{U}$ is the finite candidate set of intrinsic rewards as tasks. The goal of $\mathcal{D}$ is to minimize the regret with respect to the Pareto optimal for these objectives:

$$\max_{\mathcal{D}} \mathbb{E}_{\mathcal{D}}[\sum_{t=0}^{T} P_t^1], \; \max_{\mathcal{D}} \mathbb{E}_{\mathcal{D}}[\sum_{t=0}^{T} P_t^2], \; \cdots, \; \max_{\mathcal{D}} \mathbb{E}_{\mathcal{D}}[\sum_{t=0}^{T} P_t^p], \tag{6}$$

where $T$ is the total number of selections in the curriculum and $P$ is a vector with each entry corresponding to each evaluation metric. $p$ is the dimension of $P$ and is the number of metrics, $P^i$ is the $i$th entry of $P$. $H_t \in \mathcal{H}$ includes the $(P_i, D(H_{i-1}))$ tuples for all $i < t$. The metric $P_t$ quantifies the agent's behavior on task $\mathcal{D}(H_t)$ after a certain number of training steps.

For Empirical Pareto UCB, task is uniform randomly chosen from the Pareto action set

$$\left\{ u \mid \forall v \in U, \; \bar{\mu}_t(v) + c\sqrt{\frac{\ln(t\sqrt[4]{pK})}{N_t(v)}} \not\succ \bar{\mu}_t(u) + c\sqrt{\frac{\ln(t\sqrt[4]{pK})}{N_t(u)}} \right\}, \tag{7}$$

where $u_t \in \mathcal{U}$ is the intrinsic reward selected by UCB, and $\bar{\mu}(u)$ is the weighted sum of the past $P$ by training agent in the intrinsic rewards of the task $u$. $N(u)$ is the number of times $u$ has been chosen.

$K$ is a empirical number that upper bounds the Pareto optimal set of arms. $\not\succ$ means non-dominant. We say that $x$ is non-dominated by $y$, $y \not\succ x$, if and only if there exists at least one dimension j for which $y^j < x^j$ (Zitzler et al., 2003).

This is a nonstationary multi-arm bandit, because the intrinsic reward resulting in best learning progress might change along with the agent's learning process. There are two common ways to adapt the UCB algorithm for nonstationary situations. One way to do this is with discounting and another way is to use a sliding window (Lattimore & Szepesvári, 2020). We find that discounting has better performance in experiments. Let $\gamma \in (0, 1)$ be the discount factor, and define

$$\bar{\mu}_t^\gamma(u) = \sum_{s=0}^{t-1} \gamma^{t-s} P_s \mathbb{I}\{u_s = u\},\tag{8}$$

and

$$N_t^\gamma(u) = \sum_{s=0}^{t-1} \gamma^{t-s} \mathbb{I}\{u_s = u\}.\tag{9}$$

When using discounting for nonstationary Empirical Pareto UCB, the Pareto action set becomes

$$\left\{ u \mid \forall v \in U, \ \bar{\mu}_t^\gamma(v) + c\sqrt{\frac{\ln(\sum_{w \in \mathcal{U}} N_t^\gamma(w) \sqrt[4]{pK})}{N_t^\gamma(v)}} \not\succ \bar{\mu}_t^\gamma(u) + c\sqrt{\frac{\ln(\sum_{w \in \mathcal{U}} N_t^\gamma(w) \sqrt[4]{pK})}{N_t^\gamma(u)}} \right\}.\tag{10}$$

## 5 EXPERIMENTS

In this section, we conduct experiments to evaluate our method. We first analyze how the curriculum combines the advantages of candidate intrinsic rewards, providing intuition for its benefit in exploration and skill learning. Then, we compare the proposed method to baselines such as single intrinsic reward and random curriculum.

### 5.1 SETUP

The environments we consider are the 2D maze environments of (Campos et al., 2020), which are modified from (Trott et al., 2019).These environments are meant to provide intuitive comparison between visualizations and our proposed metrics. More experimental details and more results in high-dimensional Mujoco environments are included in Appendix. D and Appendix. F.

Baselines are in two categories:

- **Conventional URL with a single intrinsic reward**. It is also the minimal implementation of previous URL approaches.
- **Random curriculum**. We consider curriculums with randomly selected intrinsic rewards. By comparing our proposed method to this random curriculum, we could validate whether the bandit algorithm for selection is necessary.

### 5.2 ANALYSIS OF EVALUATION METRICS

Recall that in Fig. 2, there are traditional visualization results for three agents. Correspondingly, we provide evaluation results with proposed metrics in Table 1. The particle-based met-

Table 1: State Coverage in Fig. 2

|     | SC   | PMI | LSEPIN     | MSEPIN     |
|-----|------|-----|------------|------------|
| (a) | 1656 | 44  | $\approx 0$ | $\approx 0$ |
| (b) | 1386 | 881 | 145        | 275        |
| (c) | 1471 | 966 | 271        | 308        |

rics are estimated with $n = 4000$ and $k = 2000$. The state coverage of (a), (b) and (c) is $[1656, 1386, 1471]$. It is obvious that Fig. 2a has best state coverage, so this metric is consistent with the visualizations. The PMI of Fig. 2 is $[44, 881, 966]$. It is also consistent with the visualizations in which both Fig. 2b and Fig. 2c show overall informative skills. The $I(S; \mathbf{1}_z)$ for Fig.2b is {blue: 145, yellow: 509, green: 324, red: 225}. It is in accordance with the visualizations, in which yellow is the most informative skill. For the blue skill, its $H(S|\mathbf{1}_z)$ is high because it has a high $H(s|Z \neq z)$ close to $H(s)$. This is shown in visualization in which the state coverage without blue is close to the state coverage with blue. Intuitively, the blue skill is not as informative as others because it does not contribute to the total state coverage as much as other skills, so blue has a low $I(S; \mathbf{1}_z)$. The $I(S; \mathbf{1}_z)$ for Fig.2c is {blue: 345, yellow: 271, green: 258, red: 369}. The skills learned are better balanced, with red being the most informative and green slightly less informative than others. $I(S; \mathbf{1}_z)$ for every skill in Fig.2a is close to 0, because the skills are inseparable and uninformative. For the skills in Fig. 2b, the disentanglement metrics are {MSEPIN: 275, LSEPIN: 145}, and they are {MSEPIN: 308, LSEPIN: 271} for the skills in Fig. 2c. As expected from the visualizations, the skills of the agent in Fig. 2c are more disentangled than in Fig. 2b.

## 5.3 ADVANTAGES OF CURRICULUM URL

We use the two intrinsic rewards, including APT and ICM, and evaluate our curriculum with them on the tree maze environment to show how a good combination compliments each other's disadvantages. In our method, we use APT and ICM as two arms. For the tree maze environment, we consider a single objective of state coverage as defined in Section 4.2 We run APT, ICM, Random curriculum and our method with the same series of 5 random seeds. The total number of reward selections are 100.

Table 2: Two Arm State Coverage in Tree Maze. (SO indicates single objective.)

| State coverage | APT | ICM | Random | Ours (SO) |
|---|---|---|---|---|
| mean | 1397.32 | 1215.26 | 1391.09 | **1474.78** |
| std | 54.12 | 304.28 | 114.82 | **48.99** |

Table 2 shows the means and standard deviations of the state coverage. It's clear that Ours(SO) dominates both in mean performance and performance variance. The baseline of the random curriculum could not be better than the single APT, so the order of intrinsic rewards in the curriculum matters and Ours(SO) is capable of finding a good curriculum.

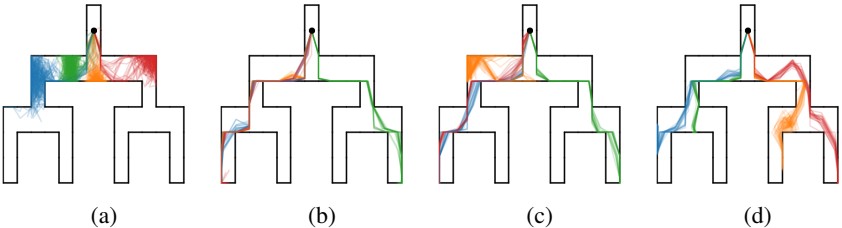

(a)          (b)          (c)          (d)

Figure 3: Tree maze trajectories of (a) APT, (b) ICM, (c) Ours (SO), and (d) Ours (MO).

By first looking at the trajectories learned by APT and ICM shown in Fig. 3a and 3b, the results show that APT prefers to make the agent expand its state coverage locally. The trajectories are getting thicker but stay near to the starting state. ICM reaches to further areas, but its trajectories are thin, possibly because its reward rely on a prediction model of environmental dynamics. When this model is accurate in large part of the state space, the intrinsic reward might lead the agent to go only along where the model is not as accurately approximated. We found that as learning progresses, Ours (SO) prefers to choose APT more. The number of times APT is chosen in the later half of training is on average 32.6% higher than the first half. This is in agreement with an intuition that for better state coverage, the agent should first reach further and then expand its trajectories.

## 5.4 Advantages of Multi-Objective for Curriculum

For the same tree maze, we consider multiple objectives: SC, PMI and LSEPIN. We use three intrinsic rewards, including SMM, ICM and APT. Table 3 shows the results that we compare the multi-objective Pareto UCB to UCB with only SC objective. Overall, MO works better than SO. It means that multiple objectives improve the agent's learning. Fig. 3 (c) and (d) show two agents learned by Ours (SO) with single objective of SC and Ours (MO) learned by multi-objective of {SC, PMI, LSEPIN} respectively. The SO agent has 3 skill on the left side of the tree, while the MO agent has trajectories of all skills separated. This is an example of why Ours (MO) has better disentanglement on average.

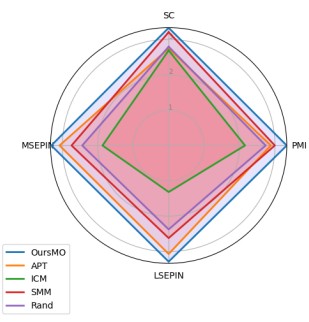

Figure 4: Multi-metrics.

Table 3: Comparison between Ours (MO) and Ours (SO). (MO indicates multiple objectives.)

|            | SC        | PMI        | LSEPIN     | MSEPIN     |
|------------|-----------|------------|------------|------------|
| Ours (MO)  | 1393.59   | **921.34** | **198.61** | **274.66** |
| Ours (SO)  | **1460.23** | 887.01   | 179.08     | 232.91     |

We also evaluate our method on larger crazy maze environment. This is an environment much more difficult to explore than the simple tree maze.

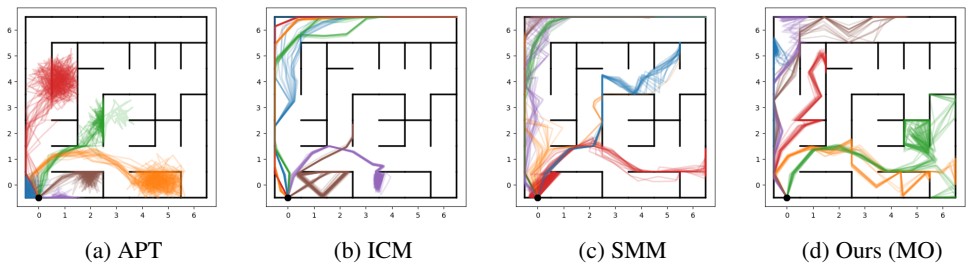

(a) APT      (b) ICM      (c) SMM      (d) Ours (MO)

Figure 5: Trajectories samples of the crazy maze

In Fig. 4, we compare our method with individual intrinsic rewards and the random curriculum baseline. As we can see from Fig. 4, our proposed method achieves a well-rounded result as expected. Also, the quantitative result intuitively accords with visualizations in Fig. 5. For example by Fig. 4, SMM has better SC metric but less disentanglement (MSEPIN and LSEPIN) than APT. In 5, SMM covered more to the upper right of the maze, but its blue and purple skills seem to be entangled with each other.

## 6 Conclusion

We proposed quantifiable and general evaluation metrics for URL. Our proposed metrics can stably measure the state coverage for exploration, as well as mutual information and disentanglement for skill learning. This helps to enable evaluation of URL without specific downstream tasks. Furthermore, we proposed an automatic curriculum to select intrinsic rewards based on the agent's learning progress. This automatic curriculum does not require prior knowledge of the environment or its intrinsic reward candidates. It is a nonstationary Pareto UCB that utilises historical evaluations for decision making, and tries to train the agent to be well-rounded at all aspects of the considered metrics. Our experimental results have demonstrated the effectiveness of our method.

ETHICAL STATEMENT

Our work does not suffer from discrimination/bias/fairness concerns.

REPRODUCIBILITY STATEMENT

The implementation code will be public after our paper is accepted.

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

## A  MODIFIED PARTICLE-BASED ENTROPY

Liu & Abbeel (2021a) find averaging the distance over all $k$ nearest neighbors leads to a more robust and stable result.

$$H_{\text{PB}}(s) := \sum_{i=1}^{n} \log \left( c + \frac{1}{k} \sum_{s_i^{(j)} \in \text{N}_k(s_i)} \| s_i - s_i^{(j)} \| \right),$$  (11)

where $\text{N}_k(\cdot)$ denotes the $k$ nearest neighbors around a particle, $c$ is a constant for numerical stability (commonly fixed to 1).

## B  APPROXIMATION OF $I(S; \mathbf{1}_z)$

It can be approximated by

$$I(S; \mathbf{1}_z) \approx \hat{H}_{\text{PB}}(S) - \hat{H}_{\text{PB}}(S|\mathbf{1}_z).$$  (12)

The second term of Equation 12 is the state entropy conditioned on knowing whether skill equals $z$, and it is defined as

$$H(S|\mathbf{1}_z) = H(S|Z = z) + H(S|Z \neq z).$$  (13)

The first term of Equation 13 can be estimated by sampling from the states generated with skill $z$. The second term is a little more tricky for parametric methods that try to approximate $\log p(s|z)$, but for particle-based entropy it can still be conveniently estimated by sampling states from all skills not equal to $z$.

## C  MOTIVATION OF DISENTANGLEMENT FROM A THEORETICAL PERSPECTIVE

Eysenbach et al. (2021) analysed that maximizing the mutual information objective $I(S, Z)$ learns skills are all optimal for some downstream state-dependent reward functions (a common family of downstream tasks). However, in practice, $I(S, Z)$ cares about only the expected informativeness of skills, only optimizing $I(S, Z)$ could result in skills that are far less than the average. By adding disentanglement metrics MESPIN and LSEPIN, we can evaluate how well individual skills are learned, in order to ensure that all skills are potentially the best skill for certain downstream tasks.

From a information geometry perspective, this paper claims that the learned skills must lie at vertices of the state marginal polytope, and for every state-dependent reward function (corresponding a downstream task), among the set of policies that maximize that reward function is one that lies at a vertex of the state marginal polytope. But the learned skills can not cover all vertices. Therefore, by adding disentanglement metrics, we care about the separatibility of skills. We want the skills to be further from each other and not to cluster only in a small region of vertices.

## D  EXPERIMENTAL DETAILS OF 2D MAZE

The experimental settings follow mainly Campos et al. (2020), and the 2D maze environments are modified from Trott et al. (2019). It is a continuous 2D environment, so the implemented RL algorithm is DDPG. The reward selection module selects a new intrinsic reward function every 500 episodes. The buffer only stores state action transitions. When DDPG update, the intrinsic rewards are calculated by the sampled states and current intrinsic reward function.

For baselines, we mentioned that we use the minimal implementation of existing URL algorithms. For example, ICM and APT can operate in the space of latent representations with additional modifications, while we implement them directly in the state space. Besides, SMM mentioned that exploration with SMM can be considered as a game between the density model of the state distribution and the agent, so fictitious play (Brown, 1951) can be implemented to help find the Nash equilibrium in finite time (Robinson, 1951). To capture this advantage of SMM, we implemented SMM with 4 historical density models.

# E MORE RESULTS FOR 2D CRAZY MAZE

Fig. 6 show sample trajectories on the the crazy maze environment, which is a more complicated 2D maze environment than the tree maze. Our method works better than others. Table 4 shows that the well-rounded performance of our proposed method is consistent.

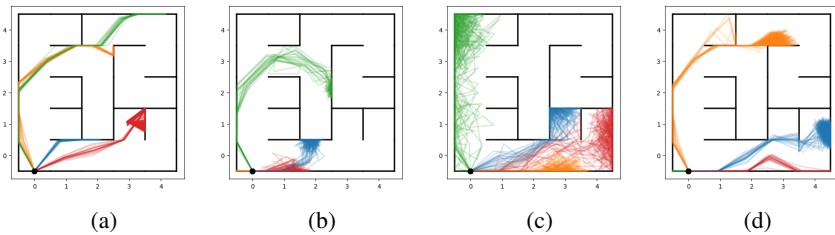

(a)        (b)        (c)        (d)

Figure 6: Trajectories samples of the crazy maze: (a) ICM, (b) SMM, (c) APT, and (d) Ours(MO).

Table 4: Comparison between Ours (MO) and baselines.

| Method | SC | PMI | LSEPIN | MSEPIN |
|---|---|---|---|---|
| Ours (MO) | **1552.81** | **1076.98** | **251.59** | **345.36** |
| Random | 1338.36 | 883.22 | 148.64 | 284.97 |
| APT | 1457.11 | 848.73 | 188.74 | 266.67 |
| SMM | 1279.97 | 764.52 | 137.35 | 229.73 |
| ICM | 1375.10 | 647.51 | 83.27 | 228.65 |

# F EXPERIMENT ON MUJOCO BENCHMARK

In the setting of URLB (Laskin et al., 2021a), the normalized performance of 4 downstream tasks in Walker domain for existing methods is shown in the radar plot Fig. 7a. We evaluated their pretrained model by our proposed metrics are shown in Table 5.

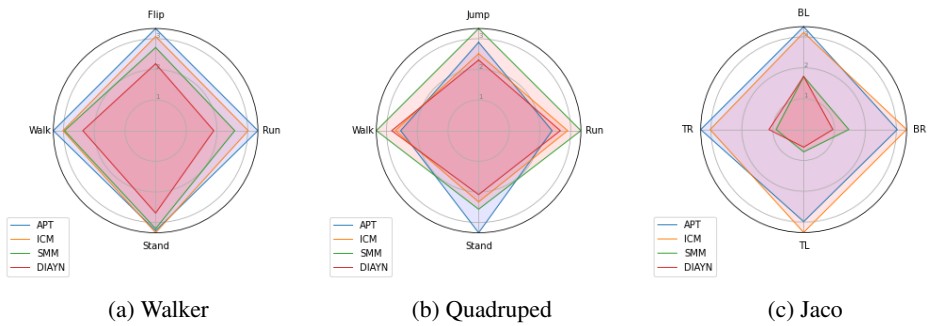

(a) Walker        (b) Quadruped        (c) Jaco

Figure 7: Radar plots for benchmark downstream performance. Each plot shows the normalized performance in each domain of 4 downstream tasks.

In URLB setting, ICM and APT only explore and do not learn skills, so they do not optimize the mutual information skill learning objective and the skills metrics are not applicable to them. The performance shown in Fig. 7 is from pretraining the agents with corresponding URL algorithm in corresponding domain for 1e5 steps.

As we can see by comparing Fig. 7 and Table 5, there is a strong correlation between overall downstream performance and the proposed state coverage metric. This shows that in more complicated

Table 5: Metrics for Benchmark domains

| Domain | Method | SC | PMI | LSEPIN | MSEPIN |
|---|---|---|---|---|---|
| Walker | ICM | 6328.43 | - | - | - |
| | APT | 6521.84 | - | - | - |
| | SMM(w MI) | 5861.81 | 179.66 | 1.24 | 60.69 |
| | DIAYN(w MI) | 4367.92 | 245.43 | 98.99 | 119.38 |
| quadruped | ICM | 6365.39 | - | - | - |
| | APT | 6357.98 | - | - | - |
| | SMM(w MI) | 6509.69 | 38.23 | 26.84 | 38.82 |
| | DIAYN(w MI) | 6135.50 | 8.55 | 7.43 | 8.66 |
| Jaco | ICM | 3202.43 | - | - | - |
| | APT | 3155.22 | - | - | - |
| | SMM(w MI) | 3663.76 | 41.16 | 13.28 | 20.59 |
| | DIAYN(w MI) | 3434.12 | 28.99 | 2.28 | 11.33 |

environments, our proposed metrics can also evaluate the properties of the pretrained agents, and these properties are correlated with downstream task performance.

A high mutual information value means that the state space is better partitioned by the skills, so the skill-conditioned policy of DIAYN with high PMI metric covers a specific region of the state space. However, these downstream tasks are not just simple partitions of the state space. Tasks such as Walk, Jump, and Run are trajectories with specific state and action sequences, so it is reasonable that the simple mutual information objective $I(S, Z)$ is not capable of assigning task-specific trajectory sequences with skills, while the high PMI metric means that the explored state space is better partitioned by skills, meaning that an individual skill-conditioned policy actually covers less state space. This explains the reason why DIAYN have a bad performance for downstream tasks, learned skills are not directly applicable to downstream tasks but they have less state coverage.

This is a problem with the objetive of DIAYN itself, and more recent URL algorithms are trying to deal with the problem of how to learn skills that are not just simple partitions of the state space, such as VISR (Hansen et al., 2020), APS (Liu & Abbeel, 2021b), LSD (Park et al., 2022), etc.

The effect mentioned above is more obvious in the domain of Jaco, where the tasks of controlling robot arm to reach BL (bottom left), BR (bottom right), TL (top left), TR (top right) have very different rewarding states, so individual skill-conditioned policy with less state coverage could have more sparse rewards at the beginning of fintuning.

In summary, our proposed metrics are capable of evaluating the properties of unsupervised RL in high-dimensional environments where visualizations are not applicable. There is a correlation between the performance of the downstream tasks and our proposed metrics, but our proposed metrics are more general and not restricted to specific tasks.

