# OpenReview forum: "AUTOMATIC CURRICULUM FOR UNSUPERVISED REIN- FORCEMENT LEARNING"
_ICLR.cc/2023/Conference — Submitted to ICLR 2023_

### Official Review · Reviewer_kRzv · 2022-10-20

**Confidence:** 3
**Correctness:** 1
**Technical Novelty And Significance:** 2
**Empirical Novelty And Significance:** 2
**Recommendation:** 1

**Clarity, Quality, Novelty And Reproducibility:**

The experimental section lacks a lot of details on the experiments, to the point that they are not, in my opinion, reproducible. For example, I cannot tell what reinforcement learning algorithm is being used, what the action space is, what the state representation looks like, if any neural networks are being used or if it is a purely tabular model and so on.

I also can't tell at what frequency the bandit algorithm is called? Does the reward change at each time-step, each episode, every couple of episodes?
Is there a different value estimation for each of the rewards or is it a joint estimation?

As a simple baseline, how would the linear combination of APT and ICM (possibly rescaled by their respective magnitude) perform?



**Strength And Weaknesses:**

The paper doesn't provide downstream evaluation of fine-tuned policies on actual task (such as, in their setting, the completion of the maze), as is commonly done in the URL literature. As such, some key underlying assumption of the paper as not empirically verified. For example, the paper proposes to maximize the combination of the proposed metrics (in a pareto optimal way). However, there is no evidence that doing so actually results in better downstream performance on tasks of interest.

One other major weakness of the work is to limit the experiments to a very toy setting, without trying to apply the findings to more challenging environments (such as Habitat, Minecraft3D, Doom, Atari) as is customary in the URL literature.




**Summary Of The Paper:**

The paper presents a multi-arm bandit method to choose between to intrinsic rewards for exploration in a synthetic maze, in an Unsupervised Reinforcement Learning (URL) setting.

It also defines some metrics to track the diversity of the policies as well as their coverage of the maze, in order to be able to compare them.

Since there are several methods of interest, the proposed method used a multi-objective bandit method.

**Summary Of The Review:**

Overall, the paper falls short of demonstrating the usefulness of the method, both in terms of downstream evaluation, as well as in non-toy settings. On top of that, the experiments lacks too many details to be replicated. I therefore recommend the rejection of this paper in its current shape.

---

> ### Author Response · Authors · 2022-11-15
> **Response to the Review (Part 1/2)**
>
> We respectfully disagree with the reviewer on several points. Here are our replies to the reviewer’s comments:
> ***
> >1. “The paper doesn't provide downstream evaluation of fine-tuned policies on actual task (such as, in their setting, the completion of the maze), as is commonly done in the URL literature. As such, some key underlying assumption of the paper as not empirically verified. For example, the paper proposes to maximize the combination of the proposed metrics (in a pareto optimal way). However, there is no evidence that doing so actually results in better downstream performance on tasks of interest.”
>
> A1: First of all, “downstream task evaluation of fine-tuned policies” is not necessary for verification of our claims. In fact, many works in URL literature relies on visualization much more than downstream task evaluation, such as [1], [2], [3], [4], [5], [6], etc. However, both evaluations have their own limitations:
> * limitations of downstream task evaluation:
>
>    1. URL is expected to learn task-agnostic knowledge (exploration and fundamental skills) that can be generally helpful to arbitrary tasks. Given the significant difference and diversity among possible downstream tasks, evaluation on a few of them can be biased or suffers from high variance, while covering all possible ones is impossible.
>
>    2. The finetuning setting of different downstream RL tasks can be very different and requires task-specific hyperparameter tuning and downstream task evaluation, which increase the variance of evaluation for URL.
>
> * limitations of visualization:
>
>    1. Visualization cannot provide a quantitative and thorough evaluation.
>    2. Though visualization might be more direct to show the results, it can be inaccurate, subjective (varying across different persons), biased, and not quantifiable.
>
> This motivates our work. Our mainly claimed contributions include:
>    1. Our proposed metrics can evaluate different preferred properties of URL in a quantifiable and task-agnostic way.
>    2. Our proposed automatic curriculum can combine advantages of multiple intrinsic rewards.
>    3. Our proposed method can optimize multiple objectives in a Pareto optimal way.
>
> * Claim 1 is supported by definition and empirically supported by Section 5.2. More support from a theoretical perspective is added in Appendix. C.
>
> * Claim 2 is empirically supported by Section 5.3.
>
> * Claim 3 is empirically supported by Section 5.4.
>
> Although improving downstream task performance is not a major focus of our paper, we added new experiments in more complicated Mujoco environments (Appendix. F). The result shows that our proposed metrics strongly correlate with the downstream task performance.
> ***
> >2. “One other major weakness of the work is to limit the experiments to a very toy setting, without trying to apply the findings to more challenging environments (such as Habitat, Minecraft3D, Doom, Atari) as is customary in the URL literature..”
>
> A2: Most URL works used 2D visualization settings for experiments and evaluation, we follow them to ensure a fair comparison with them. For a complicated maze setting, URL is also a new and challenging problem.
> * We intended to use 2D environments for intuitive comparison between visualizations and our proposed metrics.
> * To the best of our knowledge, the above-mentioned image-based environments are not “customary” in URL literature.
> * We verify in more complicated Mujoco environments that our proposed metrics faithfully reflect the downstream task performance.
>
> ***
> >3. “As a simple baseline, how would the linear combination of APT and ICM (possibly rescaled by their respective magnitude) perform?”
>
> A3: This is not a good baseline because:
> * The goal is to optimize multi-objective (maximizing both APT rewards and ICM rewards) and find Pareto optimal solutions. This cannot be guaranteed by linearization (weighted linear combination of objectives), as shown in 4.7.4 of [7]. So this “baseline” is not theoretically supported to always find an optimal trade-off between objectives.
>
> * This “simple baseline” is hard to tune and work empirically, according to our failed experiments of it. Reasons are not only limited to rescaling but also the conflicts between intrinsic rewards, which can cancel out each other after linear combination and fail.
>
> * Moreover, the combination weights strongly rely on prior knowledge, which contradicts the foundational principle of “unsupervised”.
>
> So this approach is not suitable to be included in the results.

---

> > ### Author Response · Authors · 2022-11-15
> > **Response of the Review (Part 2/2)**
> >
> > ***
> > >4. “The experimental section lacks a lot of details on the experiments, to the point that they are not, in my opinion, reproducible. For example, I cannot tell what reinforcement learning algorithm is being used, what the action space is, what the state representation looks like, if any neural networks are being used or if it is a purely tabular model and so on.”
> >
> > A4: We followed the detailed setting in previous URL works [8] since most URL methods use similar settings for visualization. The environments are continuous and neural networks are trained for policy and value approximation, which were already included in our citations from the beginning. We follow the settings of [8], so the RL algorithm is DDPG.
> >
> > ***
> > >5. “I also can't tell at what frequency the bandit algorithm is called? Does the reward change at each time-step, each episode, every couple of episodes? Is there a different value estimation for each of the rewards or is it a joint estimation?”
> >
> > A5: These details are added in Appendix. D
> >
> > ***
> > In summary, we respectfully disagree with the reviewer saying “The main claims of the paper are incorrect or not at all supported by theory or empirical results” and we sincerely suggest the reviewer to carefully examine our paper and response.
> >
> > ***
> > ## References
> > [1] Sharma A, Gu S, Levine S, et al. Dynamics-aware unsupervised discovery of skills[J]. arXiv preprint arXiv:1907.01657, 2019.
> >
> > [2] Campos V, Trott A, Xiong C, et al. Explore, discover and learn: Unsupervised discovery of state-covering skills[C]//International Conference on Machine Learning. PMLR, 2020: 1317-1327.
> >
> > [3] Lee L, Eysenbach B, Parisotto E, et al. Efficient exploration via state marginal matching[J]. arXiv preprint arXiv:1906.05274, 2019.
> >
> > [4] Kim J, Park S, Kim G. Unsupervised skill discovery with bottleneck option learning[J]. arXiv preprint arXiv:2106.14305, 2021.
> >
> > [5] Eysenbach B, Gupta A, Ibarz J, et al. Diversity is all you need: Learning skills without a reward function[J]. arXiv preprint arXiv:1802.06070, 2018.
> >
> > [6] Park S, Choi J, Kim J, et al. Lipschitz-constrained Unsupervised Skill Discovery[C]//International Conference on Learning Representations. 2021.
> >
> > [7] Boyd S, Boyd S P, Vandenberghe L. Convex optimization[M]. Cambridge university press, 2004.
> >
> > [8] Campos V, Trott A, Xiong C, et al. Explore, discover and learn: Unsupervised discovery of state-covering skills[C]//International Conference on Machine Learning. PMLR, 2020: 1317-1327.

---

> ### Author Response · Authors · 2022-11-30
> **A reminder**
>
> We understand the reviewer's load is high and we thank you again for your time!
> We just wanted to flag that we have made significant improvements to our paper, with new experiments and additional clarifications (based partly on your specific recommendations). We have also made every effort to address each of your concerns. Other reviewers have given our paper significantly higher scores, and we were hoping you might reconsider your rating.

---

### Official Review · Reviewer_sbNE · 2022-10-29

**Confidence:** 3
**Correctness:** 4
**Technical Novelty And Significance:** 2
**Empirical Novelty And Significance:** 2
**Recommendation:** 5

**Clarity, Quality, Novelty And Reproducibility:**

The paper is clearly written for the most part. However, the contributions are not clearly stated, which may contribute to an impression that the paper makes a number of more minor contributions. The novelty does not seem particularly high to me in comparison to prior work on this topic. The authors were not able to provide their code before the submission and have promised it for afterward.

**Strength And Weaknesses:**

Strengths:
- Based on my subjective view, I do believe the proposed approach has arrived at better skills in some sense.
- Each component of the system seems logical to me in the context of the goal of the paper.

Weaknesses:
- While in some sense the solution makes a lot of sense, no individual component of the system is particularly novel in light of the prior literature. Additionally, the motivation to combine all of these objectives in this way seems a bit like an over-engineered solution for a topic that is largely blue sky as currently presented and far from concrete applications.
- The proposed metrics for evaluating unsupervised RL do seem a bit arbitrary/hacky in the grand scheme of things and I feel like I get the most out of the visualizations, which is counter to the goal of this paper. Why not evaluate the skills with respect to downstream tasks? While I see where the authors are coming from arguing that the skills learned are better than the baselines visualized, it all feels quite subjective to me and really depends on what we would like to use these skills for later.
- The paper is largely based on a series of intuitions and I didn't really find a clear overarching theory that justifies the contribution.
- I worry that the proposed solution has a number of moving parts and may be sensitive to hyperparameters and particular implementation details.
- I wonder if the authors can explain more about the issue with individual objectives for unsupervised RL. It does seem counterintuitive to me that no single metric would be able to result in good skills. I just wonder if the community has not found it yet or if it is obviously something more like a meta-transfer objective. I would be much more excited about this paper if it had more to say from a theoretical perspective about this topic.

**Summary Of The Paper:**

This paper looks at the problem of unsupervised RL where skills must be learned for exploration in an environment without access to rewards. The authors take the approach of considering multiple metrics for guiding this unsupervised learning that have been proposed in the literature while choosing among each of them in an automatic curriculum learning fashion. To guide the curriculum learning process the authors use a discounted version of Pareto UCB in order to handle the non-stationary and multi-objective components of the problem. The authors compare their approach to previous baselines showcasing visualizations and metrics that may indicate their approach has intuitively learned something better or more useful.

**Summary Of The Review:**

I am a bit on the fence about this paper as I think the topic is important, the components of the proposed approach make sense individually, and the results seem pretty good. However, I feel like the novelty is quite low and the approach seems a bit over-engineered given the nature of the domains considered. As a result, I lean towards rejection at the moment, and feel that the paper could be a lot better if it engaged with more of a unified theory and discussed/explored the connection with preparation for downstream tasks more concretely.

Update After Author Feedback:

While I really appreciate the detailed response from the authors, I honestly did not find it very convincing in addressing my concerns and thus lean towards my original assessment. I will just list some of my feedback on each point listed by the authors in order to be constructive for the authors in forming subsequent revisions:

1. First of all, proving that mutual information is not an optimal single metric is very different from establishing that any such single metric does not exist. Also, the intuition that we would like each skill to be balanced in how informative it is seems purely intuitive to me.
2. This is just a restatement of the argument from the paper. My personal feeling was that the visualization was what I got the most out of.
3. While covering all possible downstream tasks may be computationally prohibitive, I don't see why it is not possible to perform a statistical analysis of performance on an unbiased sample. I appreciate the results provided for Mujoco and would suggest this kind of analysis should be further highlighted within the main text.
4. I disagree that the first point has been established. While past curriculum learning strategies may or may not be applied to unsupervised RL thus far in the literature, they are potentially applicable to unsupervised RL. Additionally, the latter points are as I stated really resulting from a combination of smaller contributions.
5. I disagree that points 1, 2, and 5 have really been established beyond reasonable doubt in this work. If these things could be formally proven, then I can see the authors perspective. However, I am currently doubtful that it is possible to show this based on the current discourse.
6. This is not what I was looking for when I asked for a theoretical justification. More concretely, I would suggest formally establishing theories related to points 1, 2, and 5 in response to the question about over-engineering that can be stated as theorems or propositions. Point 3 can be a corollary to these established theoretical statements.

---

> ### Author Response · Authors · 2022-11-15
> **A Response to the Review (Part 1/2)**
>
> We thank the reviewer for the feedback. We really appreciate your detailed and insightful feedback on our manuscript. In the following, we carefully address every concern from your comments:
>
> ## 1. About necessity of multiple metrics
> >“It does seem counterintuitive to me that no single metric would be able to result in good skills”
>
> A1: A single metric for skill learning is not enough to develop good skills because:
>
> Mutual information objective I(S,Z) is commonly used in existing unsupervised RL methods but it only evaluates the expected informativeness of skills but does not take their variance into account, e.g., the worst skill can be much less informative than the average.
>
> This motivates us to study novel metrics that evaluate the robust informativeness and separatability of skills. By considering the novel disentanglement metrics, we can evaluate the worst-case or median of the skills, which are important to balancing the learning of different skills.
>
> ## 2. About visualizations
> >“The proposed metrics for evaluating unsupervised RL do seem a bit arbitrary/hacky in the grand scheme of things and I feel like I get the most out of the visualizations, which is counter to the goal of this paper.”
>
> A2: Visualization cannot provide a quantitative and thorough evaluation.
>
> Though visualization might be more direct to show the results, it can be inaccurate, subjective (varying across different persons), biased, and not quantifiable. An important contribution of our paper (and recent work of unsupervised  RL [1]) aims to develop better evaluation metrics that can quantify different preferred properties of URL. In the paper, we report both the visualization and the metrics to provide a complete and thorough evaluation.
>
> ## 3. About downstream tasks
> > “Why not evaluate the skills with respect to downstream tasks? While I see where the authors are coming from arguing that the skills learned are better than the baselines visualized, it all feels quite subjective to me and really depends on what we would like to use these skills for later.”
>
> A3:
> * URL is expected to learn task-agnostic knowledge (exploration and fundamental skills) that can be generally helpful to arbitrary tasks. Given the significant difference and diversity among possible downstream tasks, evaluation on a few of them can be biased or suffers from high variance, while covering all possible ones is impossible.
> * The finetuning setting of different downstream RL tasks can be very different and requires task-specific hyperparameter tuning and downstream task evaluation, which increase the variance of evaluation for URL.
> * Our reported evaluation metrics can accurately reflect the performance on downstream tasks. To verify this claim, we added new experiments in more complicated Mujoco environments (Appendix. F). The result shows that our proposed metrics strongly correlate with the downstream task performance.
>
> ## 4. About contributions and novelty
> >“However, the contributions are not clearly stated, which may contribute to an impression that the paper makes a number of more minor contributions. The novelty does not seem particularly high to me in comparison to prior work on this topic. ”
>
> A4a: We respectfully disagree because:
>   1. We discovered that multi-objective optimization is necessary (single objective leads to pitfalls) and important to URL.
>   2. We proposed the first curriculum learning method for unsupervised RL (URL).
> 3. We propose novel recipe of multiple evaluation metrics that enables quantitative and thorough evaluations that for the first time can cover all major properties of URL.
>
> > “no individual component of the system is particularly novel in light of the prior literature”
>
> A4b: Besides the novelty on the addressed multi-objective URL problem and the curriculum learning framework, we also have technical novelty on proposing new individual components, e.g., the mutual information for individual skill I(S,1_z) and the disentanglement metrics MSEPIN and LSEPIN—They are not from previous works.

---

> > ### Author Response · Authors · 2022-11-15
> > **A Response to the Review (Part 2/2)**
> >
> > ## 5. About "over-engineering"
> > >“the motivation to combine all of these objectives in this way seems a bit like an over-engineered solution for a topic that is largely blue sky as currently presented and far from concrete applications”
> >
> > A5: It is not an over-engineering but addresses a fundamental problem of URL, due to the following reasoning steps (every step is well-motivated):
> >
> > 1. Single objective cannot cover all properties of URL and may lead to pitfalls. ->
> > 2. So we need to optimize multiple objectives for URL. ->
> > 3. But no existing intrinsic reward can cover all the objectives. ->
> > 4. So we should combine multiple intrinsic rewards. ->
> > 5. But trivially adding them together requires infeasible hyperparameter tuning. ->
> > 6. So we adopt one intrinsic reward per training stage and change it across different stages. This allows us to keep using previous URL implementations. ->
> > 7. For multi-objective + dynamically changing intrinsic reward, we naturally resort to a non-stationary multi-objective bandit algorithm.
> >
> > ## 6. Theoretical justification:
> >
> > A6: In [2], maximizing the mutual information objective I(S,Z) results in skills all optimal for some downstream tasks with state-dependent reward functions. However, in practice, this reduces to maximizing the averaged informativeness of skills and ignores the possibly high variance for the worst cases. By developing the new disentanglement metrics MESPIN and LSEPIN, we can evaluate how well each individual skill is learned so every skill is potentially the best skill for certain downstream tasks.
> >
> > From an information geometry perspective, this paper claims that the learned skills must lie at the vertices of the state marginal polytope. Moreover, for every state-dependent reward function (a family of downstream tasks), we can train a policy to maximize the reward function so it lies at a vertex of the state marginal polytope.  But the learned skills can not cover all vertices. Therefore, we add disentanglement metrics to improve the separability of learned skills. This keeps the skills to be further from each other and not forming clusters in a small region of vertices.
> >
> > The above theoretically motivate the multiple evaluation metrics and training objectives proposed in this paper.
> >
> > To solve the multi-objective optimization, 4.7.4 of [3] showed that optimizing the linearization (weighted sum) of objectives is incapable to find all the points in a non-convex Pareto optimal set of solutions.
> > [4] has provided the regret bound for the Pareto UCB algorithm.
> >
> > ------
> > We would like to thank you again and we hope our reply helps with your concerns.
> >
> > ## References
> > [1] Kim J, Park S, Kim G. Unsupervised skill discovery with bottleneck option learning[J]. arXiv preprint arXiv:2106.14305, 2021.
> >
> > [2] Eysenbach B, Salakhutdinov R, Levine S. The information geometry of unsupervised reinforcement learning[J]. arXiv preprint arXiv:2110.02719, 2021.
> >
> > [3] Boyd S, Boyd S P, Vandenberghe L. Convex optimization[M]. Cambridge university press, 2004.
> >
> > [4] M. M. Drugan and A. Nowe, "Designing multi-objective multi-armed bandits algorithms: A study," The 2013 International Joint Conference on Neural Networks (IJCNN), 2013, pp. 1-8, doi: 10.1109/IJCNN.2013.6707036.

---

> ### Author Response · Authors · 2022-11-30
> **A reminder**
>
> We sincerely appreciate your time and effort in reviewing our paper! We believe your constructive comments will further strengthen our paper, especially your feedback from a theoretical perspective. In the response and revised paper, we have made every effort to address your concerns. Particularly,  1.Why metrics over visualization 2.Why it is not over-engineering 3.Theoretical justification. Hence, it would be highly appreciated if you could provide feedback on our responses or confirm whether there is no remained concern. If you have any further concerns, questions, or suggestions, we are willing to discuss and reflect on them in the next revision. Thank you!

---

> > ### Comment · Reviewer_sbNE · 2022-12-05
> > **Re: A Response to the Review**
> >
> > While I really appreciate the detailed response from the authors, I honestly did not find it very convincing in addressing my concerns and thus lean towards my original assessment. I will just list some of my feedback on each point listed by the authors in order to be constructive for the authors in forming subsequent revisions:
> >
> > 1. First of all, proving that mutual information is not an optimal single metric is very different from establishing that any such single metric does not exist. Also, the intuition that we would like each skill to be balanced in how informative it is seems purely intuitive to me.
> > 2. This is just a restatement of the argument from the paper. My personal feeling was that the visualization was what I got the most out of.
> > 3. While covering all possible downstream tasks may be computationally prohibitive, I don't see why it is not possible to perform a statistical analysis of performance on an unbiased sample. I appreciate the results provided for Mujoco and would suggest this kind of analysis should be further highlighted within the main text.
> > 4. I disagree that the first point has been established. While past curriculum learning strategies may or may not be applied to unsupervised RL thus far in the literature, they are potentially applicable to unsupervised RL. Additionally, the latter points are as I stated really resulting from a combination of smaller contributions.
> > 5. I disagree that points 1, 2, and 5 have really been established beyond reasonable doubt in this work. If these things could be formally proven, then I can see the authors perspective. However, I am currently doubtful that it is possible to show this based on the current discourse.
> > 6. This is not what I was looking for when I asked for a theoretical justification. More concretely, I would suggest formally establishing theories related to points 1, 2, and 5 in response to the question about over-engineering that can be stated as theorems or propositions. Point 3 can be a corollary to these established theoretical statements.

---

> > > ### Author Response · Authors · 2022-12-10
> > > **2nd Response to the Review**
> > >
> > > Thanks again on your comments. We really appreciate your focus on theories, which is valuable in today’s AI research community.
> > >
> > > It seems that all the concerns about novelty and over-engineering comes down to a theoretical analysis of the necessity of multiple metrics.
> > >
> > > Here is a sketch of what our theorem would be like and how we are going to prove it. We will include it in our future work:
> > >
> > > The theorem would state that when the mutual information objective is maximized, a higher LSEPIN metric prepares the skills for better initialization for the worst case. A sketch of proof would be like this:
> > > 1. In the setting of [1], previous results have shown when I(S, Z) is maximized, the learned skills are on the vertices of the feasible state marginal polytope, and they are concyclic in a sense that they have equal KL divergence to the average state marginal distribution, so they are on the circle with average state distribution p(s) being the center.
> > > 2. We don’t assume the vertices are non-concyclic, so there could be different learning skills maximizing I(S, Z).
> > > 3. It will be proved first that when a skill z is known, for any desired state distribution that is furthest from z (D(z||m) is highest among the learned skills), the adaptation cost (minimum KL divergence other skills chosen from the vertices on the circle to the desired state distribution) is upper bounded. And this upper bound can be lowered by increasing I(S, I_z)
> > > 4. With the necessity of I(S, I_z), comes the necessity of LSEPIN.
> > >
> > > This theorem will be a major contribution for us to be the first to formally prove the necessity of disentanglement metrics for unsupervised skill learning, although the motivation for disentanglement metrics is already very intuitive.
> > >
> > >
> > > Here is the response for each of your new concerns:
> > > 1. It is an empirical problem existing in previous URL work and also shown in our paper that the skills can be unbalanced by learning with the default option of mutual information for URL. The necessity of the disentanglement metrics can not only be based on theoretical justification but also based on this shortage of empirical results in previous research.
> > >
> > > 2. A major shortage of visualizations is that it is only at most two or three dimensions. When you try to lower the dimensions for visualization, you need prior knowledge of the importance of dimensions, which contrasts the concept of unsupervised.
> > >
> > > 3. Even if it is unbiased samples, it is still expensive to evaluate a large number of downstream tasks for an unbiased evaluation, so it will suffer from high variance. Besides, you can not have an unbiased sample without knowing what task is possible for the environment, which means that you need prior knowledge of the environment (eg. the reachable trajectories) and it contrasts with the concept of “unsupervised”. What’s more, downstream task evaluation can not be performed online during the learning process, which is also a shortage compared to our proposed metrics.
> > >
> > > 4. You can argue that the technical contributions are small, but if you can agree on the necessity of multiple metrics after reading our new response and theoretical proof, then you should also agree that the whole system is a major conceptual contribution. The simple fact is that we are the first to use automatic curriculum learning for URL for good reason.
> > >
> > > 5. For point 5 in response to the question about over-engineering, there is no need for us to prove this general multi-objective optimization issue.
> > > As we mention in 6 of “A Response to the Review”:
> > > “4.7.4 of [3] showed that optimizing the linearization (weighted sum) of objectives is incapable to find all the points in a non-convex Pareto optimal set of solutions. [4] has provided the regret bound for the Pareto UCB algorithm. ”
> > > So it all comes down to points 1 and 2 about the necessity of multiple metrics and multi-objectives, which will be theoretically justified in the future.
> > >
> > > 6. We will include the formal theoretical justification later. This analysis also inspires us to further look into the theoretical justification for skill-learning methods with other probability distances. [1] and our analysis both use KL divergence, which is non-symmetric and doesn’t satisfy the triangle inequality. When we make the skill learning as an objective of max E_z D(p(s|z), p(z)) with a true distance D, the analysis would be much simpler and could lead to much more interesting results.
> > >
> > >
> > >
> > >
> > >
> > >
> > > [1] Eysenbach B, Salakhutdinov R, Levine S. The information geometry of unsupervised reinforcement learning[J]. arXiv preprint arXiv:2110.02719, 2021.

---

### Official Review · Reviewer_eCNr · 2022-11-04

**Confidence:** 3
**Correctness:** 4
**Technical Novelty And Significance:** 4
**Empirical Novelty And Significance:** Not applicable
**Recommendation:** 6

**Clarity, Quality, Novelty And Reproducibility:**

The paper has great clarity. The proposed approach is novel, and the experiment settings are clear.

**Strength And Weaknesses:**

Strength:
* The paper is well-written and easy to understand.
* The approach is novel and well motivated.
* The experimental results are clearly showcased.

Weaknesses:
* The experiments are only based on a single 2D maze environment, which makes the results not convincing enough. There are other more realistic environments to further test on, e.g., https://arxiv.org/pdf/2110.15191.pdf.
* There is no results to show if the proposed unsupervised RL method can really help in downstream applications, although the proposed approach doesn't require assumptions on specific downstream task.

**Summary Of The Paper:**

For unsupervised reinforcement learning, this paper proposes (1) a set of metrics to evaluate the exploration and skill learning of the agent without specific downstream tasks; (2) an automatic curriculum to train the agent with different intrinsic rewards in different stages, formulated by multi-armed bandit.

**Summary Of The Review:**

The paper is well-written and the proposed approach is novel, but the experiments are only based on a single 2D environment makes the results not fully convincing. Besides, as an unsupervised approach, it's not tested in any downstream applications.

---

> ### Author Response · Authors · 2022-11-15
> **A Response to the Review**
>
> We thank the reviewer for the feedback. We really appreciate that you understood our motivations and strengths very well and provided us with insightful comments. We carefully address every concern from your review:
>
>
> >1. “The experiments are only based on a single 2D maze environment, which makes the results not convincing enough.”
>
> A1: We have added experiments for the Mujoco environments URLB [1] and more complicated maze environments in our modified manuscript.
>
> >2. “There is no results to show if the proposed unsupervised RL method can really help in downstream applications, although the proposed approach doesn't require assumptions on specific downstream task.”
>
> A2: Yes, we do not rely on downstream tasks because we argue that downstream task evaluation has the following limitations:
>
> * URL is expected to learn task-agnostic knowledge (exploration and fundamental skills) that can be generally helpful to arbitrary tasks. Given the significant difference and diversity among possible downstream tasks, evaluation on a few of them can be biased or suffers from high variance, while covering all possible ones is impossible.
> * The finetuning setting of different downstream RL tasks can be very different and requires task-specific hyperparameter tuning and downstream task evaluation, which increase the variance of evaluation for URL.
>
> Our reported evaluation metrics can accurately reflect the performance of downstream tasks. To verify this claim, we added new experiments in more complicated Mujoco environments (Appendix. F). The result shows that our proposed metrics strongly correlate with the downstream task performance. Because of this correlation, our proposed method, which aims to optimize multiple metrics to find a desirable Pareto optimal solution, can potentially prepare the unsupervised reinforcement learning agents for better downstream tasks adaptation.
>
>
> ## References
> [1] Laskin M, Yarats D, Liu H, et al. URLB: Unsupervised reinforcement learning benchmark[J]. arXiv preprint arXiv:2110.15191, 2021.

---

> ### Author Response · Authors · 2022-11-30
> **A reminder**
>
> Thanks again for your time and your valuable comments. We have addressed your concerns and made significant improvements to our paper, with new experiments and additional clarifications, especially about the connections between our proposed metrics and downstream performance. Besides, we have added discussions from a theoretical perspective. Hence, it would be highly appreciated if you could provide new feedback and potentially raise the score, given you were originally "borderline accept" and our paper is now much stronger.

---

### Author Response · Authors · 2022-11-30
**Summary of changes**

In the revision, we have:
1. Added more experimental results, especially Mujoco environments (used in URLB [1]) to show the connection between our proposed metrics and the downstream task performance
2. Added clarifications and discussions from a theoretical perspective about the necessity of proposed metrics.
3. Added more experimental details.





[1] Laskin M, Yarats D, Liu H, et al. URLB: Unsupervised reinforcement learning benchmark[J]. arXiv preprint arXiv:2110.15191, 2021.

---

### Decision · Program_Chairs · 2023-01-20

**Decision:**

Reject

**Justification For Why Not Higher Score:**

The experiment mainly on 2D maze environment tends to be toyish, and evaluation through visualization is subjective, and the approach seems a combination of existing components that lack strong novelty.

**Justification For Why Not Lower Score:**

The problem is interesting and the paper is well written. The approach achieves certain improvement in learning skills.

**Metareview: Summary, Strengths And Weaknesses:**

The paper studies unsupervised RL that learns skills for exploration an envorinment without specific rewards. The proposed approach integrates multiple metrics (in a pareto optimal way) and choosing among each of them with automatic curriculum learning. The approach is compared against previous baselines with visualization and other metrics. As the reviewers pointed out, the experiment mainly on 2D maze environment tends to be toyish, and evaluation through visualization is subjective, and the approach seems a combination of existing components that lack strong novelty.